# Review on Conductive Polymer/CNTs Nanocomposites Based Flexible and Stretchable Strain and Pressure Sensors

**DOI:** 10.3390/s21020341

**Published:** 2021-01-06

**Authors:** Olfa Kanoun, Ayda Bouhamed, Rajarajan Ramalingame, Jose Roberto Bautista-Quijano, Dhivakar Rajendran, Ammar Al-Hamry

**Affiliations:** Professorship of Measurement and Sensor Technology, Chemnitz University of Technology, 09111 Chemnitz, Germany; rajarajan.ramalingame@etit.tu-chemnitz.de (R.R.); roberto.bautista@etit.tu-chemnitz.de (J.R.B.-Q.); dhivakar.rajendran@etit.tu-chemnitz.de (D.R.); ammar.al-hamry@etit.tu-chemnitz.de (A.A.-H.)

**Keywords:** polymer/CNTs nanocomposites, strain sensors, polymer/CNTs nanocomposites pressure sensors, piezoresistive, piezocapacitive, stretchability

## Abstract

In the last decade, significant developments of flexible and stretchable force sensors have been witnessed in order to satisfy the demand of several applications in robotic, prosthetics, wearables and structural health monitoring bringing decisive advantages due to their manifold customizability, easy integration and outstanding performance in terms of sensor properties and low-cost realization. In this paper, we review current advances in this field with a special focus on polymer/carbon nanotubes (CNTs) based sensors. Based on the electrical properties of polymer/CNTs nanocomposite, we explain underlying principles for pressure and strain sensors. We highlight the influence of the manufacturing processes on the achieved sensing properties and the manifold possibilities to realize sensors using different shapes, dimensions and measurement procedures. After an intensive review of the realized sensor performances in terms of sensitivity, stretchability, stability and durability, we describe perspectives and provide novel trends for future developments in this intriguing field.

## 1. Introduction

Nowadays, there is a growing demand for flexible and stretchable strain and pressure sensors in several applications, such as structural health monitoring [1], human motion detection [2,3], robotic touch applications [4], skin-mountable sensors [5,6], and several further applications [6,7] (Figure 1). In fact, conventional sensors like common metallic foils and rigid semiconductors have shown several limitations, such as low sensitivity and low flexibility and stretchability (usually with ε < 5%), which make them incompatible with different structure shapes [8,9] and not applicable in a wide range of emerging applications. The stretchability refers thereby to the ability of the device to regain the original shape after deformation. Therefore, stretchability is considered as new requirement of flexible electronics especially for applications such as skin-mountable or robotic touch applications. 

To date, numerous works reported on new approaches for fabrication of flexible, stretchable and sensitive force sensors. They are mainly based on the use of sensitive materials coated on a flexible substrate [10] or a stretchable sensitive material [11]. These sensitive materials can realize sensors having different principles, such as piezoresistive [12,13], piezoelectric [14], and capacitive [15] principles. Nanocomposite piezoresistive and piezocapacitive materials provide a good alternative to conventional materials owing to their higher sensitivity, easy manufacturing [16]. Especially, materials based on polymer nanocomposites containing smart carbon nanomaterials, such as graphene [16], carbon nanotubes (CNTs) [17,18,19,20] gained tremendous attractivity in the realization of pressure and strain sensors due to their superior sensitivity, their outstanding mechanical and electrical properties, in addition to the possibility of tunability of the material properties of the nanocomposite, which can be realized in respect to the desired application requirements. Particularly, polymer/CNTs composites have enhanced mechanical and electrical properties in comparison to the other nanofillers. In fact, CNTs have a large aspect ratio (diameters ranging from 1 to 100 nm and lengths of up to mm) and a conjugated structure promoting a ballistic charge transport. Hence, electrical conductivity of composite is enhanced at a very low concentration in comparison to other conductive fillers (Figure 2a,b).

In fact, spherical particles such as carbon blacks possess low aspect ratio compared to CNTs because of the large diameter making the formation of conductive paths very difficult at low particle concentration not as in the case of CNTs.

Realization of wearable pressure and strain sensors based on polymer/CNTs with outstanding performance including superior stretchability, good flexibility, a wide sensing range, and sensitivity requires the consideration of several aspects such as the used polymer, CNTs size and orientation, adopted processing approach and sensor structure. 

In general, elastomers are mainly used to offer the sensor superior stretchability and flexibility. The commonly used polymers for wide strain range are thermoplastic polyurethane (TPU) [21], ecoflex [22], and poly(dimethylsiloxane) (PDMS) [23]. These materials offer the possibility to shape sensor in different geometries which widen their application range. 

Until now, previous review papers on polymer/CNTs strain and pressure sensor are mainly addressing the electrical properties of nanocomposite and their influencing factors [24,25]. However, no reports are illustrating the possibility of polymer/CNTs to endow both piezoresisitive and piezocapacitive characteristics and showing possible structure forms of polymer/CNTs strain and pressure sensors and their potential applications.

In this review, we report on recent developments in flexible and stretchable polymer/CNTs films realizing high performance strain and pressure sensors. The paper is organized as follows: In Section 2, we introduce the sensing principles for pressure and strain sensors based on polymer/CNTs nanocomposite materials and consider thereby the factors responsible for improving the general properties of nanocomposite materials. In Section 3, we explore the fabrication process of polymer/CNTs materials. In Section 4, we focus on the sensor development for strain and pressure measurement and consider thereby the influence of fabrication techniques on the resulting sensor properties, such as sensitivity, linearity, stability as well the potential applications for developed polymer/CNTs sensors. Section 5 includes on one hand the novel trends towards polymer/CNTs for physical sensors and on the other hand the potential of polymer/CNTs for chemical sensors. Section 6 is a summary of the main results within the review.

## 2. Sensing Principles for Pressure and Strain Sensors

### 2.1. Basic Phenomena of Percolation 

The combination of CNTs with polymer leads to radical change of electrical conductivity. An increase of the electrical conductivity with several orders of magnitude larger than that of neat polymer owing to the construction of 3D electrically conductive networks. In fact, the transition from insulator to conductor by the addition of CNTs follow the percolation theory caused from large disparity between the electrical conductivity of the polymer matrix and CNTs. Therefore, as demonstrated in Figure 2 three transition phases are seen. This theory is represented by the following equation: *σ* = *σ*_0_ (*ρ* − *ρ*_c_)^*t*^ for *ρ* > *ρ*_c_,(1)
where *σ*_0_ is the physical parameter attributed to the intrinsic conductivity of CNTs and *t* is the critical exponent which is dominated by the dimensionality of the system. According to this theory, it exists at a critical concentration of CNTs namely percolation threshold (*ρ*_c_) at which electrical conductivity increases suddenly. 

The resistance of a CNT network consists of two of resistances; intrinsic resistance 𝑅_𝑡𝑢𝑏𝑒_ of the CNT and the is inter-tube resistance 𝑅_𝑗𝑢𝑛𝑐𝑡𝑖𝑜_*_n_*. The latter can be either the contact resistance *R_C_* of CNTs in physical contact, e.g., ohmic contact to metallic electrode or tunneling resistance *R_T_* of CNTs separated by a small gap e.g., polymer or a combination of both resistances. At low concentrations region, the electrical conductivity has a non-linear increase by slightly increasing CNTs concentrations up to a certain concentration. The magnitude of change is low because CNTs are completely integrated into the polymer and less contacts between them exist. Therefore, the conduction mechanism is contributed via quantum tunneling, where the tunneling resistance is orders of magnitudes larger than the resistance of direct CNT contacts. The overall resistance is influenced by several parameters such as polymer dielectric properties, the tunneling gap *D* between adjacent nanotubes and the electronic structure of the nanotubes. At percolation conductive paths begin to be built allowing the current to pass through the CNTs, which are becoming more in contact together. In the region of CNT concentrations beyond the percolation threshold, the conductive filler network follows more the classical geometrical percolation theory. Extensive experimental studies have studied the percolation of the composite and its dependency on several factors as summarized in Table 1. These factors are mainly the dispersion preparation process, CNTs aspect ratio, CNTs orientation, CNTs state, polymer viscosity, etc.

Geng et al. [34] found that electrical properties of the nanocomposite depend on the CNTs state. It was found that treating the CNTs with two different surfactants i.e., triton X-100 and silane modification leads to changes of the final electrical conductivity of the nanocomposite. Triton X-100 was found to be more effective on the dispersion of carbon nanotubes in polymer than silane chemical treatment due to the formation of strong interface between the CNTs and epoxy matrix introduced by the hydrophobic and hydrophilic segments of the nonionic surfactant. The orientation of CNTs can lead to lower the percolation threshold and can therefore reduce the amount of nanofillers necessary to realize a good sensitivity. For example, Du et al. [35] examined the effect of alignment of Single wall CNTs (SWNCT) in a PMMA polymer on the electrical conductivity of the composite. They employed melt fiber spinning and the extensional flow in the spinning process and adjusted thereby the alignment level. They demonstrated a.o. that higher conductivities can be realized by slight alignment of the CNTs. 

The polymer properties are critical and lead to significant change on the electrical conductivity. Especially, low polymer viscosity is important for several manufacturing steps and helps to achieve a low percolation threshold. In this regard, several studies applied solvents to lower the viscosity. For example, Thostenson et al. [36] lowered the percolation threshold down to 0.1 wt.% of CNTs and reached this by the addition of 40 wt.% styrene to the vinyl ester monomer reduces the viscosity. The percolation threshold was thereby five times lower than the classical vinyl ester monomer, which is a significant improvement.

### 2.2. Piezoresistive Effect 

Piezoresistive effect refers to changes of geometry or/and resistivity of a material with strain. In the case of CNTs/polymer nanocomposite change significantly their resistivity and provides therefore a great potential to be used in various interesting applications. The CNT/polymer nanocomposites display inverse behavior relying on the applied strain; a negative piezoresistance effect (NPRE) under pressure where the resistance of nanocomposites decreases with an increase in pressure stress and a positive piezoresistance effect (PPRE) under tensile load where the resistance of nanocomposites increases with an increase in tensile stress. 

The behavior of CNTs/polymer nanocomposites can be attributed to inverse mechanisms as depicted in Figure 3. In the case of tensile stress, mainly following phenomena take place: (a) Decrease of CNTs conductive baths due to loss of CNTs contact becoming more tunneling-dominated ones, (b) tunneling conduction mechanism influencing conductivity change in adjacent CNTs due to gap variation between CNTs and (c) intrinsic conductivity changes of CNTs due to deformation. Application of strain leads therefore to bending and rearrangement of CNTs in the host polymer, which results in the initialization of new paths and destruction of some others. 

In the case of tensile stress, the tunneling effect is the most dominant effect affecting piezoresistivity regardless the CNT concentration, while in pressure stress, the loss of tunneling effect and formation of new paths are the most dominant effects affecting piezoresistivity. For evaluation of piezoresistive behavior of CNTs/polymer nanocomposite, the sensitivity or gauge factor (GF) can be calculated using the following formula.
*GF = εΔR/R*_0,_(2)
where *∆*𝑅 is the change of the sensor resistance, 𝑅_0_ initial resistance at 𝜀_0_ = 0. 𝛥𝑙 is the change of length, and 𝑙_0_ is the nominal length. 𝜀 is the applied strain. 

According to Equation (2), higher sensitivity can be achieved when a large resistance change is obtained by small external load. For nanocomposites, piezoresistive sensitivity attains usually the maximal value near the percolation threshold. To get higher sensitivity to a tensile load, it is recommended to be at the percolation threshold because the application of a little external load can induce a significant change on the network due to the tunneling effects. The mechanical properties of the composite do not change significantly above the percolation threshold. The high concentration of nanofillers leads to the formation of a hard composite and makes it not capable to deform easily under an external load. While in pressure, it is recommended to work under or approximately near the percolation thresholds in order to lose remarkably the tunneling effects.

### 2.3. Piezocapacitive Effect

Polymer-carbon nanocomposite based piezocapacitive sensors can be broadly classified into two distinct categories depending on their electrode structure. One being the parallel plate structures where two conductive electrodes are separated by a dielectric spacer that is either insulating or conducting depending on the material composition. The other being an interdigital electrode structure that is in combination with a conductive sensing layer. While principally both structures can measure the sensor response as a change in capacitance under applied pressure, the later has advantage or the other with its simple, cost-effective fabrication technique and the ability to sense an extended pressure range [37]. 

Piezocapacitive sensors based on parallel plate electrode structures work on the principle of change in capacitance as the area between the parallel plate changes under shear force or the distance between the parallel plate changes under perpendicular force. The capacitance of such sensors can be easily computed using the relation C= EA/d, E  where is the dielectric constant of the material between the parallel plate electrodes, *A* is the area and *d* is the distance between the parallel plate electrodes as represented in Figure 4a [38]. These sensors are highly suitable for measurement of small forces with high sensitivity due to the highly flexible soft elastomer that comprise the dielectric layer. Theses sensors work with both insulating and conducting dielectric layer. However, the sensors based on interdigital electrode structures always require a conductive layer that is placed on top of the electrode layer. The capacitance of such a sensor is a parallel combination of individual sensor capacitance between each electrode pair. Upon application of pressure the change in capacitance of the sensor is a result of integral change in individual capacitance of each electrode pair. This change in capacitance inherently comes from the change in tunneling capacitance between the CNTs as they come close to each other under compression and the movement of CNTs towards the electric field distribution of the underlying electrodes as represented in Figure 4b [37].

## 3. Polymer/CNTs Nanocomposite Materials 

Manifold polymer nanocomposites can be produced by the combination of a polymer matrix and a variety of nanofillers. These nanofillers can have one type, of two- or three-dimensional like CNTs or graphene materials. Particularly, CNTs are promising due to their structural and functional properties such as high aspect ratio, high mechanical strength and outstanding electrical properties. Therefore, embedding CNTs in different polymer matrices can realize materials with attractive and customizable properties. A big challenge thereby is that CNTs tend to be attracted to each other caused by the high Van der Waals forces leading to form large clusters or agglomerates. Hence, the performance of the nanocomposites is highly affected by the distribution of CNTs in polymers. Therefore, many efforts have been devoted to realize a homogeneous CNTs dispersion within the polymer matrix. This is indispensable for realizing sensors and builds the basis for a good reproducibility. 

In order to tailor and enhance the dispersion of the CNTs/polymer composites, many approaches have been adopted, such as direct mixing, solution mixing, melt processing and in situ-polymerization, which are detailed in the next sections.

### 3.1. Fabrication Approaches

#### 3.1.1. Solution Processing

Solution processing is one of the most common and simplest approach for processing CNT/polymer nanocomposites. In general, CNTs are dispersed into a suitable solvent by stirring, mixing or sonication applying mechanical energy to unbundle the CNTs. The CNTs dispersion is then mixed with polymer followed by controlled evaporation process. An efficient CNTs distribution can be already achieved in the first step using sonication process. Sonication can be in two forms, by bath or tip. In general, this process is based on the application of high energies to disperse solid powder-like form of the CNTs into liquid solutions. Indeed, the horn sonicator is more efficient to exfoliate CNTs in reasonable time comparing to the bath sonicator due to high power that can be delivered. Using a tip sonicator, it is possible to control the quality of dispersion by careful selection of the intensity and time. In fact, excessive sonication time or amplitude leads to undesired side effects [39]. Therefore, it should be carefully used to avoid fracture of CNTs, which can occur and change thereby the percolation ratio as well as the film properties.

#### 3.1.2. Melt Processing

The main difficulty in dispersing CNT agglomerates is overcoming the high intermolecular van der Waals interactions and entanglements between CNTs. However, proper dispersion is necessary to obtain benefits from the CNT when fabricating polymer composites. Melt processing is used as alternative to solution processing methods in order to solve the incompatibility and solubility problem of the solvent with the polymer matrix. This method is usually used for thermoplastic materials as the polymer is melted to form a viscous liquid and then CNTs is blended with the melted polymer. However, it should be kept in mind that polymer weight degradation and also CNT shortening can occur due to the high shear forces applied. During melt processing, it is expected that the shear stresses transferred from the polymer melt onto the primary CNT agglomerates promotes dispersion and distribution of the particles into the polymer [40].

On one hand, the efficiency of melt mixing methods is typically lower in comparison to solution processing methods due to limitation by the high viscosity, which inhibits the uniform dispersion of CNTs [41]. For instance, Sandler et al. [42] has found that depending on the processing conditions and CNTs type, the electric percolation threshold of CNT composites can vary from 0.001 wt.% to 10 wt.% depending on the mixing methods (solution, dry or melt mixing) and the type of polymer employed (thermoplastic or thermoset) [40,41]. Thus, the processing technique including the dispersion conditions of the CNTs in the matrix is a crucial factor determining the electrical conductivity of the composite, which in turns affects the sensing behavior of CNT/polymer composites. On the other hand, considering sensor fabrication melt processing is considered to be the most viable option for fabricating polymer composites based on thermoplastics matrixes towards large scale production for industrial applications given its low cost and flexibility [43]. Moreover, Jouni et al. [41] found for high density polyethylene (HDPE) that electric percolations as low as 0.30 vol% can be achieved in melt processes by diluting a master batch combined with melt mixing. This shows the possibility to achieve suitable electrical properties in thermoplastic polymers for sensing applications using melt processes. Additionally, awide range of available technologies of well-established melt compounding is applicable to polymer/CNT composite fabrication [40]. Therefore, melt processing can be considered as the most affordable method for polymer nanocomposites strain sensors fabrication in terms of large-scale fabrication and industrial applications.

#### 3.1.3. In-Situ Polymerization

In order to overcome the aforementioned dispersion issues between CNTs and polymer, it has been proposed to fabricate CNT composites via in-situ polymerization and is also suitable for polymers that are insoluble and thermally unstable and cannot be prepared by solution or melts processing [44]. This process is based on the dispersion of CNTs into monomer matrix with or without the presence of solvent, where a standard method of polymerization is then performed [45]. The main advantage of this technique is that dispersion can be improved if the CNTs are provided with functional groups compatible with the monomer [46,47]. However, during synthesis an insulating polymer layer is formed on the CNTs surface hindering the tunneling mechanism within the composite [48]. Thus, the strain sensitivity could also be negatively affected by the formation of a tunneling barrier during in-situ polymerization. 

#### 3.1.4. Functionalization of CNTs

Despite of various techniques involved in the fabrication polymer/CNTs nanocomposites, it is still challenging to achieve a repeatable, precise and economical nanocomposite fabrication [49,50]. It was reported that functionalization of CNTs can enhance the homogenous dispersion of the conductive nanofiller in non-conductive polymer. Azizighannad et al. [51] fabricated the functionalized CNTs/PDMS pressure sensor and reported that functionalization of CNT can enhance the piezo-resistive behavior of the nanocomposite which is due to homogenous distribution of the CNTs and better interfacial interaction between CNTs and the polymer. This is because of electrostatic repulsion of the functional groups on the CNTs which in turn increase in the interface adhesion between the polymers and maintain the CNTs in exfoliation state. This results in enhancement of mechanical properties such as initial modulus and thermal degradation compared to the pristine CNTs. Mcclory et al. [52] analyzed the distribution behavior of functionalized CNTs by the rheological properties of CNTs/PMMA nanocomposites which shows that it has capability of forming high rheological percolated network in PMMA. Even though excellent dispersion with enhanced mechanical properties is achieved, the addition of functional groups will degrade the electrical properties of the nanocomposites. This due to disturbance in the graphite structure of nanotubes by introducing the sp^3^ hybridization inside the nanotubes and also simultaneous loss of pi -conjugation system on graphene layer [53]. This sp^3^ hybridization can increase the electronic bandgap and also seen as defect sites in terms of electron transport thorough CNT and results in reduction of reduction of tunneling mechanism [53]. It was also noted that high interface adhesion of the functionalized CNT between the non-conductive polymer which will impede the flow of electron between the CNTs i.e., hopping mechanism. Chemical functionalization is based on the covalent linkage of functional entities onto CNT surfaces [53,54]. 

Covalent functionalization of CNTs

Covalent functionalization of CNTs involves the altering the connectivity status of bonds in the carbon atoms present in the CNTs [55]. In this method, the translational symmetry of the CNTs and graphene is disrupted by changing sp^2^ carbon atoms to sp^3^ carbon atoms, and the properties of nanofillers, including both the electronic and transport properties, are influenced. The functional units form the covalent linkage with graphene skeleton or CNTs. There are several approaches such as radical polymerization, click chemistry, biofunctionalization and metal nanohybrization where it depends on the functional groups added on the CNTs [55].

Non-covalent functionalization of CNTs

Non-covalent functionalization is attaching the functional groups on CNT based on supramolecular complexation using various adsorption forces, such as Van der Waals force, hydrogen bonds, electrostatic force and π-stacking interactions, compared to other methods it could be carried out on the relatively mild reactions conditions [56]. The disadvantage of non-covalent functionalization is that it destroys the perfect structure of CNTs, resulting in considerable changes in their physical properties [56]. H-bonding and pi–pi stacking play an important role in noncovalent functionalization and improve solubility and assembly without pi–pi conjugation of the CNT or graphene skeleton. Several approaches such as polymer wrapping, surface adsorption and endohedral method are based on non-covalent functionalization technique. However, the interfacial interaction of non-covalent functionalized CNTs with polymer is weak compared to covalent functionalization CNTs [56]. 

### 3.2. Influence of Fabrication Approaches on the Electric Properties of the Material

In addition, Socher et al. [57] demonstrated that the electrical properties of nanocomposite significantly rely on original viscosity of the polymer. Big differences in the electrical percolation thresholds were found using the same polymer matrix having different viscosities, where lower viscosity is favorable for low percolation threshold. It was remarked in this study that for low viscosity polymer, more particles per area were detected, but with smaller size in comparison to the high viscosity polymer. Therefore, the rupture of larger agglomerates is more advanced in systems based on a low viscosity matrix due to the higher input of mixing energy.

According to Tjong et al. [58] polypropylene (PP)/multiwalled carbon nanotube (MWNT) nanocomposites prepared via melt-compounding exhibits much lower percolation threshold when it is prepared at higher shear rates. Therefore, higher shear rates are favorable to ensure the uniformity of the CNTs dispersion. In another study, researchers have illustrated that the shear rate can affect the orientation of CNTs and fosters the electrical conductivity at one direction and at low CNTs concentrations [59]. However, some other works show that higher shear rates can damage CNTs and destroy the contact between CNTs [60,61].

The influence of fabrication process parameters was also deeply studied in the work of Hu et al. [62]. In this study, several parameters such as curing, mixing speed and time, solvent addition, and hardener addition timing have been investigated during fabrication for their influence on the electrical properties of MWCNT/epoxy nanocomposites prepared by an in-situ polymerization process. The curing temperature and the mixing conditions are key factors in the fabrication process. High temperature during the curing process has significant impact on the conductivity since the formation of macroscopic conducting network may be formed more easily.

## 4. Development of Flexible and Stretchable Strain and Pressure Sensors Based on Polymer/CNTs Nanocomposite

### 4.1. Flexible and Stretchable Polymer/CNTs Strain Sensors Fabrication Methods and Performances

A diversity of composite mode and structure have been developed to produce flexible and stretchable polymer/CNTs composites for strain sensing. Based on the polymer/CNT composite modes and structure, three main configurations or architectures as illustrated in Figure 5 can be found: elastomer or flexible polymer filled with randomly oriented CNTs, flexible polymer composite with oriented CNTs and sandwiched structures. Depending on the architecture, strain sensors will possess different sensing performances that is suitable for a specific application. The preparation processes of each architecture will be described in the next part with illustrating the main factor responsible on the change of sensor characteristics as shown in Table 1.

#### 4.1.1. Flexible and Stretchable Strain Sensors Fabrication Methods 

Flexible polymer filled with randomly oriented CNTs

1. Flexible thin film

Usually polymer/CNT nanocomposite strain sensors are prepared by directly dispersing CNTs into a flexible polymer matrix by melt compounding, solution, or direct mixing, as discussed earlier in Section 3.1. For example, Bouhamed et al. [17] developed flexible nanocomposite strain sensor based on simple and cost-effective methods, which is direct mixing and printing using the stencil printing method as shown in Figure 6a. In order to guarantee a good distribution of the CNTs fibers within the epoxy matrix, the influence of the processing parameters was studied in this work. Based on this study, a highly sensitive strain sensor is achieved around 14 at very low CNTs concentration around 0.3 wt.%. According to this study, excessive mixing can contribute in the shortening and fracturing of CNTs leading to less conductive path formation and re-agglomeration of short CNTs which in turn can affect the sensor performance including the reproducibility and stability over time.

In Sanli et al. [63] flexible thin film strain sensors have been developed based on epoxy with randomly distributed MWCNTs. After optimization of processing parameters, a gauge factor of 78 could be realized with a with a small creep of ±0.26% and a fast response and recovery time as well as a low hysteresis under cyclic tensile and compressive loadings. These properties of the nanocomposite sensors are much better than those of classical metallic strain gauges and others are at least similar, what the direct comparison carried out in this paper shows explicitly.

2. Filament composite

Unlike other material morphologies, filaments have the advantage of being able to be woven or embroidered into textiles and in terms of strain sensing they are highly isotropic (highly sensitive in the filament direction). Moreover, if such filament sensors are made of highly stretchable polymers the flexibility and adaptability of such sensors allows their integration on several types of fabrics in any desired direction. 

Therefore, highly stretchable conductive composite filaments have attracted interest of researchers as they provide an effective way to obtain wearable strain sensors. The fabrication processes to obtain polymer composite filament sensors are mainly based on wet/dry extrusion methods as shown in Figure 6a. It is worth mentioning that for these techniques it is required to have CNTs previously dispersed, either as solution (wet extrusion), as a master-batch (dry extrusion) or in the case of melt-extrusion the CNTs are mixed during the extrusion process.

Several filaments based on polymer/CNTs have been developed where the most common polymer matrix are thermoplastic elastomers such as thermoplastic polyurethane (TPU). For example, He et al. [64] prepared a highly sensitive and stretchable wearable strain sensor based on MWCNT/TPU fiber obtained via wet spinning process. In this work, a DMF-MWCNT/TPU suspension was extruded from a syringe through a 23 G (inner diameter: 340 μm) spinneret into an acetone-based coagulation bath, and the continuous wet fiber was collected and immersed in hot water to remove the residual solvents and surfactants from the filament. In their work, He et al. reported a gauge factor (GF) as high as ~2800 in the strain range of 5–100%, while for the strain range of 0–5% the GF was ~550. The very high sensitivity reported by He et al. can be attributed to the combination of the processing method selected and the attainment of highly oriented CNTs from the syringe extrusion, such phenomena have been reported to occur in extrusion processes with small diameters die holes [65].

Tang et al. [66] developed a core−sheath fiber (CSF) via co-extrusion wet-spinning for wearable strain sensors. sore: MWCNT/Ecoflex (a silicone elastomer) MWCNT content 2 wt.%; sheaths: neat Ecoflex. Highly stretchable strain sensors up to 600% stretching were made., It has exhibited different GFs at different regions of stretching starting from GF = −0.063 (about 25% critical strain) to a maximum GF of 1378 at maximum stretching of 330%. At higher CNT concentrations (3%), the strain sensor showed lower GF = −0.45 (110% strain), up to GF = 153 at maximum stretching of 600%. During quasi-transient step testing (up to 100% of strain), the sensors exhibited overshooting in response to acceleration followed by relaxation back to a plateau value. However, upon relaxation the strain sensors did not recover to their original resistance values due to destruction of CNT network paths. Another coaxial strain sensor was fabricated by Zhou et al. [67] using wet spinning, core: SWNCTs; sheath: TPE. Up to 250%, a residual strain of 10–15% was visible (hysteresis). The performance of the filament sensor showed two linear regions (for strain from 0% to 5%, GF = 48 and linearity of 0.99, and for strain from 20% to 100%, GF = 425 linearity of 0.98. Additionally, CNT/acrylonitrile butadiene styrene (ABS) composite (CNT content 6 wt.%) were melt blended at 190 °C to form starting composite materials for filament fabrication which was prepared by laboratory scale extruder at 220 °C [68]. The CNT conductive network paths in the filament sensors can be reformed during creep and cycling tests along with continuous decrease of gauge factor that eventually stabilizes at about 2.5. However, according to temperature evaluation tests such composite fused filament fabrication (FFF)-made filaments shown stable sensing behavior in the range −25 °C and +60 °C.

Torres et al. [69] prepared melt-extruded filaments made of conductive master batches of TPU (containing 15 wt.% of MWCNT) and neat SBS. A two peak response visible for strains up to 2.5% (at 1.8 mm/min speed). However, a repeatable response was observed upon cyclic tests. Similar Quijano et al. [70] melt-extruded filaments at a SBS/C-TPU composition of 7/3, a linear behavior was observed for elongations up to ~35% of strain and a steady electrical resistance response can be obtained for elongations up to 50%, while changes were measurable up to 150% of strain; for these filaments a gage factor of 26 was obtained. Such filaments have been already proven by Rajendran et al. [71] to have potential in wearables by attaching them to a textile sensing glove capable of detection finger motion in real-time with application in hand injuries rehabilitation 

From all the above-mentioned strain sensing filaments, the simplest and most cost-effective alternative towards large scale fabrication are thermoplastic elastomers fabricated by melt-extrusion. This method can provide meters of sensor elements that are highly stretchable, easily embroidered into textiles, in a continuous one-step fabrication process. Thus, melt-extrusion of nanocomposite filament sensors are a very promising alternative towards multipurpose wearable strain sensing technology. Furthermore, depending on the diameter and rheological properties of these nanocomposite filaments it is possible to 3D print structures using CNT-polymer filaments as raw material. For instance, in a recent work Mora et al. Showed that electrically conductive polylactic acid (PLA) and HDPE filled with CNT can be used as feeding material for 3D printed composites via fused filament fabrication [72]. Moreover, Shi et al. [73] prepared conductive nanocomposite parts that were 3D-printed from PLA/CNT filaments via local enrichment strategy. Using a commercially available 3D-printers, Gnanasekaran et al. [74] obtained printed parts from filaments of polybutylene terephtalate (PBT)/CNT and PBT with graphene via classic fused deposition modeling (FDM). Therefore, conductive filament nanocomposites with sensing properties opens up a new possibility of 3D-printing sensors with more complex shapes and geometries.

Flexible polymer filled with oriented CNTs

Apart from using polymer filled with randomly arranged CNTs, a lot of works are working on aligning the CNTs within the polymer matrix in order to enlarge the strain sensing capabilities. Suzuki et al. [75] developed a strain sensor based on aligned CNTs where they stacked a dry-spinnable MWCNT web prepared by fabricated by chloride-mediated chemical vapor deposition (CM-CVD). and sandwiched between elastomer layers. The sensor was stretched up to 200% with a gauge factor above 10 and rather has a short sensing delay of fast response time below 15 ms.

Another work, Ryu et al. [76], reported about the fabrication of highly stretchable wearable sensor fabricated from dry-spun CNTs fibers. By introducing a preliminary strain of 100% to the highly oriented CNT fibers spun on stretched on a polymer (Ecoflex), the elastic performance of the strain is improved and has the ability to be stretched by over 900% while retaining high sensitivity, responsiveness, and durability. Based on the same process, a biaxial strain sensor can be also fabricated by biaxially oriented CNT fiber arrays. The biaxial sensor exhibits independent cross sensitivity. These devices can be integrated for wide application such as the detection of animal and human motion. 

Liang et al. [77] developed ultra-stretchable strain sensor based on the integration of forest of CNTs prepared by growing CNTs on quart plates using the chemical vapor deposition (CVD) method. Then, the CNTs film were placed into the mold and poured of with Ecoflex rubber as shown in Figure 6b. In this work, a comparison of this structure was done with two other structures that one contains isotropic CNTs or randomly oriented CNTs and the other contains gradient structure CNTs synthesized with the same process. This study demonstrates a great potential of the gradient structure CNTs due to the combination of randomly oriented and well aligned CNTs, acting as sensitive and stretchable conductive elements. This kind of sensor achieve high sensitivity (gauge factor (GF) = 13.5) and ultra-stretchability (>550%). The sensor great potential for applications in health monitoring, sports performance monitoring and soft robotics. 

Flexible sandwiched composite with CNTs

Another configuration is used in the literature in order to realize transparent and wearable strain sensor that can mimic human movements. This configuration consists of integrating of the CNTs thin layer between two layers of stretchable polymer. Chen et al. [78] reported about a simple spray coating and transfer method to fabricate a stretchable and transparent strain sensor based on stacked PDMS/CNTs/PDMS nanocomposite having ultrathin and uniform conductive layer as depicted in Figure 6c. The prepared strain sensors exhibit up to 250% strain, as high as 60% optical transparency, and excellent sensing performances. Stretching range up to 130% such as in human joints can be carried out with rather good stability and repeatability. 

Sui et al. [79] developed aligned SWCNTs film prepared using sliding coating method and then this layer was sandwiched between PDMS layers by mold casting in Petri dish. The developed sensor exhibits different performance depending on the loading direction. An initial crack detection was found at strain of 15% along the ⊥ direction and 12% along the // direction.

#### 4.1.2. Flexible and Stretchable Strain Sensors Performances

Table 2 summarizes a literature survey about the recent developments in flexible and stretchable strain sensor based on CNT/polymer composite taking into account the type of polymer, fabrication process, the achieved performance and the structure of the sensor. Regarding the sensitivities of the prepared strain sensors, it is clear from Table 2 that the nanocomposites present higher sensitivity than the metallic strain sensor and can achieve up to 117,213 depending on the used polymer and the CNTs filler orientation and fabrication process. According to this table, the alignment of CNTs within the polymer matrix leads to better performance in terms of sensitivity and repeatability as well wider strain range. Several other numerical works illustrate that CNTs orientation affect enormously the strain sensor performance [80,81] where high sensitivity can be achieved when CNTs are aligned to the direction of loading.

The main challenging thing during realization of aligned CNTs nanocomposite is the complexity of the preparation process in comparison to randomly oriented CNTs nanocomposite. 

In general, the alignment of CNTs can be done through electrospinning method or CVD process or post dispersion process using dielectrophoresis effect [81] or application of high magnetic field [82,83].

However, sandwiched structure shows the simplest fabrication process among the different strain sensor configurations with acceptable performance. In fact, the conductivity of this structure is mainly formed from direct connection of individual CNTs. Here, the sensor working principle is based on the loss of contact between neighboring CNTs. This means that sensors having sandwiched structure are not dominated by tunneling effect between CNTs. This latter will conduct to achieve more linear sensing response but with less sensitivity to strain as well to environmental changes [84].

Regarding the filament-based strain sensors, it is clearly visible that the processing method have a significant influence in the sensing properties of the filament. While the filaments manufactured by FFF have a GF of 3.5 up to 4% of strain, the filaments made by CWS showed a GF up to 1378 for a maximum strain of 330%. In addition, temperature has a significant influence in CWS made filaments making them difficult to use in applications where temperature cannot be controlled and changes in the temperature are expected. Furthermore, the strain range of the filaments shows to be generally larger than any other kind of composite sensors reaching up to 600% for elastomer-based filaments. Interestingly, filament sensors show to require the application of certain number of initial cycles before their response becomes repeatable, pointing out the need of a “training” process before their implementation as strain sensors. Moreover, as reported by the authors the flexibility of strain sensing filaments and their adaptability makes them to be highly suitable for wearable applications as shown in Figure 6d. 

Furthermore, the developed nanocomposite materials have the potential to be used for realization of multifunctional sensor. Several works show the cross sensitivity of polymer nanocomposite to other quantities such as solvent or gas, temperature, humidity and IR illumination. 

In addition, Table 2 demonstrates that almost of strain sensors are mainly based on piezoresistive working principles. While few works are working in the direction of capacitive strain sensor using polymer/CNTs. In fact, capacitive strain sensors based on polymer/CNTs require integration of more CNTs within the nanocomposite materials which leads to non-cost-effective solutions. 

**Table 2 sensors-21-00341-t002:** Flexible and stretchable polymer/CNTs strain sensors performances, PR: piezoresistive, PC: piezocapacitive.

Type	Materials	Fabrication Process	Strain (%)	Gauge Factor	Sensing Principle	Repeatability	Application	Cross- Sensitivity	Response Time	Ref.
Thin film-based strain sensor	Aligned CNT/Polycarbonate-urethane (PCU)	Dry-spinnable MWCNT array by chloride-mediated CVD method	500	10	PR	180,000 Cycles	Wearable, real time, human body motion sensing	-	15 ms	[75]
	(poly-vinylpyrrolidone) Polyurethane/MWCNT laminate MWCNTs oxidized with KMnO4	Electrospinning	300	450	PR	1000 Cycles	Human breath monitoring	Solvent vapor sensing capability	-	[85]
	5.46 vol%. MWCNTs in OBC (elastomeric ethylene-α-octene block copolymer)	Melt mixing	300	Randomly CNTs: 5.46 Aligned CNTs: 248	PR	Not tested	Human motion detection	Not tested	-	[86]
	Carbon nanotube/thermoplastic polyurethane (CNT/TPU) nanocomposites	3D printing by fused deposition modeling (FDM), and 1-pyrenecarboxylic acid (PCA)	-	11,7213	PR	Up to 1000 Cycle	Not tested	Not tested	-	[87]
	6% MWCNT/HEPCP nanocomposite	Solvent mixing	40–340	Increased by 421.47 time at 340% strain	PR	Not tested	e-skin and wearable devices	IR illumination and temperature sensing capabilities	-	[88]
	0.48% CNTs modified by silane coupling agent (SCA)	Swelling/permeating method	350	20	PR	Not tested	Flexible sensor field	Not tested	-	[89]
	Silicon lamina: Dragon skin/CNTs	-	up to 300%	resolution < 1%	PC	10,000 cycles at 100% strains	Human motion detection Prototypical data glove and respiration monitor	temperature sensitivity of −0.13%/°C	100 ms	[90]
	MWNT/PDMS Ecoflex/MWCNTs	Blending method	120% 300%	-	PC	-	e-skin application	Not tested	-	[91]
Filaments strain sensors	Coaxial structure, sheath: TPE, core: SWCNT	Coaxial wet spinning	100%	GF = 48 for ε < 50%	PR	Up to 3250 cycles	Expansion SHM and wearables	Not tested	<1 s	[67]
	Coaxial structure, sheath: Ecoflex, Core: Ecoflex/MWCNT	Coaxial wet-spinning (CWS)	Up to 600%	GF = −0.063 for ε = 0–25%, GR = 0.68 for ε = 50–100%, GF = 1378 for ε = 330%	PR	Up to 10,000 cycles	Wearables	Temperature: −80% change in R0 for T = 100 °C (R_ref_ taken at 0 °C)	<295 ms	[66]
	Acrylonitrile-butadiene-styrene/MWCNT	Fused filament Fabrication (FFF)	<4%	GF = 3.5 for ε = 3%	PR	Fairly repeatable only after 40 cycles for 10 cycles.	SHM	Not tested	~1 s	[69]
	MWCNT-TPU/SBS	Melt extrusion	~150%	GF = 26 for ε = 0–50%	PR	Repeatable after the 5th cycle	Wearables and sports	Not tested	~1 s	[70]

### 4.2. Flexible and Stretchable Polymer/CNTs Pressure Sensors Fabrication Methods and Performances

#### 4.2.1. Pressure Sensors Fabrication Processes

Parallel plate structure with insulating dielectric layer

The design of a parallel plate structure with two conductive electrodes separated by an insulating dielectric material is the most common and well-known technique to realize a pressure sensor as illustrated in Figure 7a. These sensors work on the principle of both piezoresistive and piezocapacitive but predominantly piezocapacitive utilizing the change in distance between the electrodes under application of pressure as a change in capacitance of the sensor. Several researches are being carried out in this direction of pressure sensors, with the focus to obtain high sensitivity to pressure by modifying the electrode/dielectric material and the geometry of the sensor.

One approach proposed by Pekarek et al. [92] as illustrated in Figure 7b, is to design a pressure sensor using a combination of a solid electrode and an elastic electrode-sensing film developed by wet anisotropic etching. The electrode comprises a highly doped silicon substrate and MWCNTs arrays grown on top of the silicon substrate grown by plasma enhanced chemical vapor deposition (PECVD) by using iron as a catalyst to increase the surface area. SIMAX glass acts as the insulating dielectric layer between the electrodes constituting the pressure sensor. Similarly, Shao et al. [93] demonstrates the use of MWCNT as a conductive material for the electrodes of a parallel plate capacitive pressure sensor. The electrodes are made flexible using utilizing polydimethylsiloxane (PDMS) as a substrate and coating it will a solution of MWCNTs dispersed in alcohol by ultrasonication. A 20 nm thick layer of gold is sputtered over the MWCNTs to enhance the conductivity of the electrodes. Two such electrode layers are sandwiched between a 1 µm thick layer of parylene C deposited through CVD on the surface of the MWCNTs, acting as the dielectric layer. The complete structure is encapsulated with a thin layer of parylene C through CVD. Yet another cost-effective fabrication of flexible capacitive pressure sensor as proposed by Maddipatla et al. [94] is to screen print a patterned top electrode and a complete bottom electrode on either side of a PDMS substrate with conductive CNT ink. As PDMS has a better adhesion property the CNT ink can be directly printed with any addition adhesion promoting agent or surface treatment, resulting in a simple fabrication technique to realize a parallel plate capacitive pressure sensor as illustrated in Figure 7c. However, Woo et al. [95] proposed lithography and pattern transfer technique to realize a layer-by-layer stacked structure of a matrix of parallel plate capacitive pressure sensors as illustrated in Figure 7d. The approach involves the preparation of a conductive material CPDMS where 10 wt.% of CNTs are dispersed in toluene solvent and mixed in PDMS to fabricate the conductive electrodes of the sensor while the standard PDMS material serves as the dielectric layer. Similarly, Cagatav et al. [96] proposed a lithographic approach to pattern the dielectric layer composed of PDMS as micropillar of 100 µm in diameter. This microstructured dielectric spacer is sandwiched between two Kapton substrates spray-coated with CNT dispersed in sodium dodecyl sulphate (SDS) delivering 100 Ω/sq, constituting the pressure sensor.

Parallel plate structures with conductive dielectric layer

The sensor principle of parallel plate structures with conductive dielectric layer is like that of Section 4.2.1 with the advantage of having a tunable conductive dielectric layer by the introduction of CNTs. Addition of CNTs in the dielectric layer enhances its conductivity facilitating realization of pressure sensors that could work on either piezoresistive or piezocapacitive principles or both.

Zhao and Bai [13] illustrated the synthesis of a conductive composite based on 0.64 vol% of graphite nanoplatelet–CNT hybrids synthesized by CVD and dispersed in PDMS through mechanical blending process. The composite is fabricated as a disc-shape with 10 mm diameter and 3 mm thick. Platinum electrodes are attached to both sides of the disc to form a piezoresistive pressure sensor as illustrated in Figure 8a. Another research proposed by Chen et al. [97] deals with synthesis of a conductive material by dispersing 0.5 wt.% CNT in liquid crystal (LC-CNT) using ultrasonication. The fabrication of the sensor structure follows a layer-by-layer photolithographic approach starting with the coating of a parylene layer on a Si wafer followed by sputtering of Au film (200 nm) which serves as the bottom electrode. The hydrogen liquid is prepared in a specific combination of chemical in a certain proportion and the hydrogen layer is fabricated using a copper mold baked at 80 °C for 30 min. Oxygen plasma treatment is utilized to bond the hydrogen layer with the Au electrode layer. The LC-CNT is then injected into the circular pattern followed by the sealing of the top electrode layer with a combination of hydrogen layer bod to the Au layer. The entire structure that comprises of a 4 × 4 matrix of piezocapacitive pressure sensors is supported by a Kapton tape for better protection as illustrated in Figure 8b. 

Yoon et al. [98] proposed an interesting approach on utilizing ion-gel as electrodes for a capacitive pressure sensor. The conductive material comprises of CNTs dispersed in toluene at a concentration of 1 mg/mL using bath sonication for 2 h, then mixed with PDMS-toluene solution by mechanical stirring. The conductive layer is synthesized by spin-coating the conductive material and curing on a chloromethlysilane coated sandpaper mold for easy detachment. Another layer of CPC with the same technique is fabricated and sealed together to form the dielectric spacer. Two electrode layers are synthesized using ion-gel and bonded to the CPC layer, copper tapes are attached to the ion-gel electrodes for electrical contacts. The complete structure is encapsulated with a thin layer of PDMS. A quite different approach is proposed by Tripathi et al. [99], where a combination of MWCNTs and ZnO are used to fabricate the conductive layer. MWCNTs are grown on Si substrate using CVD and ferrocene as catalyst followed by a uniform growth of ZnO microspheres on MWCNTs by dispersing zinc acetate dehydrate into a mixture of pre-dispersed MWCNTs in N,N-dimethlymethanamide (DMF) using ultrasonication for 30 min. After a vigorous stirring for 5 h at 105 °C the solution is transferred as a flexible paper (200 µm) of ZnO-MWCNTs using vacuum filtration technique. Copper tapes are attached on either side of the ZnO-MWCNT paper followed by encapsulation with a thin layer of PDMS. Yet another interesting approach that utilizes the combination of ZnO and MWCNTs is proposed by Li et al. [100], where the ZnO nanoparticles are used as a sacrificial layer. The nanoporous piezoresistive composite is prepared by mixing uncured PDMS with ZnO nanoparticles and MWCNTs. The MWCNTs would entangle with the ZnO nanoparticles due to the Van der Waals force. After curing material, the ZnO nanoparticles are removed by etching with hydrochloric acid, resulting in a nanoporous conductive layer, which is sandwiched between conductive tapes acting as electrodes. 

However, research on utilizing the synthesis of CNT-PDMS conductive layer to fabricate a piezoresistive pressure sensor is proposed by Miao et al. [101]. To synthesis the conductive material, CNTs are dispersed in toluene solution and then mixed vigorously with PDMS for 4 h. The curing agent is added, and the mixture is heated at 80 °C for 15 min to evaporate the toluene solvent. A PDMS monolayer wrinkle structure mold is prepared by subjecting the uncured PDMS solution to a C4F8 plasma and then curing at 80 °C for 30 min. The conductive CNT-PDMS material is powered on the wrinkle PDMS mold and peeled off to form the conductive layer. The easy release of the conductive layer is facilitated by the fluorocarbon layer on the surface of the wrinkled mold. Polyethylene terephthalate (PET) substrates with indium tin oxide (ITO) layer are attached on either as electrodes and the structure is encapsulated with PDMS. Mitrakos et al. [12] proposed a simple method to improve the resolution of piezoresistive pressure sensors at lower pressures. The method involves micro structuring of aluminum electrodes to realize truncated pyramids of 300 µm height for the bottom electrode layer. The conductive piezoresistive film is synthesized by dispersing 1 wt.% MWCNT in toluene solution by ultrasonication followed by mixing PDMS/toluene solution to the dispersed. Further ultrasonication and mechanical stirring ensures good dispersion of the MWCNT in PDMS, a solution is heated to 100 °C to evaporate the toluene solvent and then the curing agent is added. After the curing process the piezoresistive layer is attached to plain aluminum top electrode using epoxy conductive silver paste, copper wires are also attached in the similar manner to both top and bottom electrode. The complete is assembled as encapsulated using a thin polyimide tape as illustrated in Figure 8c.

Interdigital electrode structure with conductive sensing layer

The sensors in this category are a cost-effective and easy to construct solution as only two functional layers are required to fabricate the sensor, a conductive piezoresistive or piezocapacitive layer and an underlying electrode layer with interdigitated electrodes as shown in Figure 9a. 

Yogeswaran et al. [102] demonstrates a concept to realize piezoresistive pressure sensor by screen printing CNT-PDMS conductive nanocomposite on interdigital electrode structure. The interdigital electrode structures are fabricated by e-beam evaporation of chromium (10 nm) and silver (100 nm) using a 3D printed shadow mask on an oxygen plasma treated 80 µm thick PDMS substrate spin coated on a silicon wafer. The conductive composite material is synthesized by dispersing 7 wt.% MWCNT in PDMS using mechanical mixing for 10 min and ultrasonication for 30 min. The curing agent is added in the ratio 30:1 to PDMS to enhance the stretchability of the layer and is deposited on the fabricated interdigital electrode structure by screen printing technique with 3D printed mask. The sensor layer is cured in the oven at 80 °C for 2 h to bonding it to the underlying electrodes. A similar research is presented by Sethumadhavan et al. [103] where both the underlying electrode structure and the sensitive composite layer are fabricated by screen printing technique. A 125 µm PET is used as the substrate to screen print interdigital electrode structures using a mesh with thread diameter of 48 µm. The sensitive material comprises of two components: a polymer paste and a conductive carbon paste which are mixed together. Different polymers like polyvinyl alcohol (PVOH), polydimethylsiloxane (PDMS) and polyvinyl acetate (PVA) are dispersed in different solvents like isopropyl alcohol and toluene to form a diluted solution which is then mixed with carbon paste at 25%, 50% and 75% loading. The carbon paste is synthesized by dispersing carbon flakes of diameter 3–4 µm in cyclohexane by manual stirring for 7 min. The conductive carbon-polymer paste is then screen printed on top of the interdigital electrodes and cured at 120 °C for 10 min.

Further research is conducted in the similar direction by Lee et al. [104], where MWCNTs are used to synthesize the conductive layer instead of the carbon flakes. The underlying electrode layer is a copper interdigital electrode structure on FR4 substrate. The conductive material is synthesized by dispersing 15 wt.% of MWCNT in toluene using ultrasonication and mixing this dispersed in a solution of PDMS/toluene using a centrifugal mixer. The mixture is then heated to evaporate the toluene followed by adding the curing agent and material in painted on a PDMS substrate of 3.5 mm thick which is then transferred on to the electrode structure. A systematic approach on synthesizing MWCNT/PDMS conductive nanocomposite at a lower concentration of MWCNT and a significantly better performing pressure sensor is demonstrated by Ramalingame et al. [37]. The sensitive layer is synthesized by an optimized dispersion process that involves a combination of ultrasonication and magnetic stirring to disperse 0.3 wt.% of MWCNTs in THF organic solvent. PDMS is directly introduced into the dispersion and the combination of ultrasonication and magnetic stirring is repeated. After adding the curing agent, the composite is then transferred into a laser cut Teflon mold with circular slots of 200 mm in diameter and 500 µm in thickness. The material is then degassed for 15 min and gradually heated to facilitate simultaneous curing of the material and evaporation of solvent. The final thickness of the cured sample is approximately 400 µm thick and is placed on a circular interdigital electrode structure screen printed with silver nanoparticle ink on a 125 µm PET substrate as shown in Figure 9b. The sticky nature of the material enables self-adhesion to the electrode substate.

#### 4.2.2. Performance of the Pressure Sensor

The polymer-carbon nanocomposite-based pressure sensors work on the principle of either piezoresistive or piezocapacitive behavior. Such pressure sensors are characterized by key performance indices of such as the pressure sensitivity, the pressure range, the ability to detect small forces and withstand larger forces. The performance of these sensors is greater influenced by the concentration of CNT in the sensitive layer, which in turn governs the pressure sensing range, leading to various applications that require shuttle to higher forces.

The conductive filler of pressure sensor is usually comprised of a binary material system, one being the insulating polymer matrix and the other the conductive nanofillers like CNTs. The conductivity in the polymer matrix is established by dispersing CNTs in a certain concentration. The optimum concentration of CNT filler in the polymer matrix to establish a desired conductivity is crucial, as this governs the performance of the sensor. The resistance of the composite material could change vary in the range of several megaohms to few kiloohms depending on the different CNT concentration loading starting from 0.25 wt.% to 2.0 wt.% [102]. Such an investigation is essential to determine the percolation threshold of the composite which could be different for different sensors depending on the type of nanofiller, polymer, sensor geometry and sensor structure. A notable advantage of using CNTs as nanofillers is the higher aspect ratios of CNTs enabling to obtain the percolation threshold at much lower concentrations which can be observed in Figure 10a. In addition to this the quality and method of dispersing CNTs in polymer matrix also greatly alters the percolation point, facilitating conductivity at a significantly lower concentration of 0.75 wt.% [103] compared to the reported 7 wt.% [13]. The same investigation can be applied for piezocapacitive pressure sensors to determine the optimum pressure performance of the sensor. As piezocapacitive pressure sensors rely on the dielectric constant of the sensitive layer rather than the conductivity a further reduction in the CNT concentration is observed in such sensors starting from 0.2 wt.% [97]. However, an optimum pressure sensitive was obtained at a concentration of 0.5 wt.% as shown in Figure 10b. 

Though concentration of CNTs in the polymer matrix plays a vital role in tuning the performance of the sensor, the sensor structure, geometry and measurement principle is yet another important aspect that governs the overall pressure sensitivity and pressure range of the sensor. Complex parallel plate structures were realized with the aim to obtain higher sensitivity in a wide range of applied pressure. Such sensor structures mainly focus on patterning the dielectric layer by different technique like lithographic patterning of micro pillars in PDMS, sandwiching the dielectric layer as two parts and deposition of PDMS in a wrinkled mold. Though all approaches present a good sensitivity and pressure range, the structures with insulating dielectric material that utilizes CNTs as the electrode material shows sensitivity only in a small range such as 20% in 0.01–1 N [96] and 1.33 kPa^−1^ in 0–758 Pa [93]. While the sensor with a CNT-PDMS conductive layer exhibits pressure sensitivity comparatively higher-pressure range at 8.3 kPa^−1^ in 0–3.5 kPa [12]. However, with a combination of ion-gel electrodes and structured CNT-PDMS conductive layer resulted in a higher-pressure range with a sensitivity of 9.55 kPa^−1^ in 0–8 kPa.

Further enhancement in both the sensitivity and range is demonstrated by selective electrode design structure yet maintaining the same parallel plate structure concept. Through screen printing of defined electrode shapes with CNT ink and utilizing gold as the electrode layer for a liquid-crystal/CNT conductive layer higher pressure ranges were reached with a sensitivity of 0.021% kPa^−1^ in 15.2–337 kPa [94] and maximum sensing range of 400 kPa [98]. The combination of micromachined aluminum electrode for punctual pressure measurement and conductive sensitive layer lead to highest pressure sensitivity of 1.2 MPa [12] which is also realized with a better linearity in the sensor response using Ecoflex as dielectric material and CNT-PDMS as electrode layers [95].

On the other hand, simple sensor structures that are considerable easy to fabricate are designed based on the interdigital electrode concept. As the conductive sensing layer is the key for such sensor structures, optimizing the material composition of the conductive layer can unveil pressure sensitivity in further higher-pressure ranges. Research has proven that even with this promising technology the choice of nanofiller is crucial to design the pressure sensitivity, as it could cost a significantly higher concentration of nanofiller to obtain measurable conductivity thus resulting in a poor sensitivity to pressure [103]. However, CNTs as a nanofiller material has proven its potential in both establishing a good conductivity at significantly lower concentrations and enhancing the sensitivity and the pressure range. CNT-PDMS based conductive material were realized by painting and screen-printing technique obtaining an operating range of 1.25–43.75 kPa [104] and 0–1.4 N [12] respectively. In the synthesize of CNT-PDMS conductive layer, the dispersion of CNT in the polymer matrix directly alters the behavior of the sensor. With an optimized dispersion procedure, CNTs can be effectively and homogeneous distributed in PDMS matrix at a lower concentration of 1 wt.% to achieve sensitivities as high as 46.8%/N in the range 0–1 N with a minimum sensing of 1 g weight and a maximum sensing range of 180 N (~18 kg) [37]. Figure 11 shows the piezoresistive [105] and piezocapacitive [37] behavior of the CNT-PDMS pressure sensor. A detailed performance comparison is presented in Table 3.

Research has foreseen several interesting applications of these flexible polymer-carbon nanocomposite-based pressure sensors for both human and robot. Specific applications include touch and grasping applications where a stretchable nanocomposite matrix is wrapped around the finger to track the bending of fingers and to estimate the grasping force at fingertips to hold an egg [13]. A more dynamic arrangement of pressure sensors in matrix layout is shown in Figure 12a as a potential grasping application to calibrate the grasping force of robot fingers with the ability present pressure distribution in a real time visual interface. A similar 4 × 4 matrix layout concept is shown in Figure 12b, with the aim to serve as a mattress for decubitus monitoring in bedridden patients with a minimum weight resolution of 20 g [105]. The soft nature of the polymer leads to thin sensor structures that are flexible enough to detect physiological signals like movement of muscles caused by a gesture and pulse when attached to the human body as shown in Figure 12c This enables detection of gestures that can be potentially translated into a machine-readable signal to facilitate human-machine interface for robot control and telemanipulation. On the contrary, these polymer sensors are also capable of withstand higher forces as demonstrated in Figure 12d, where the sensor is using to monitor the gait of a humanoid robot [37]. The sensor is based on interdigital structure concept and effectively monitor the walking pattern of a humanoid robot which could facilitate robot’s posture maintenance, self-calibration and provide realistic diagnostic data on the gait of the robot.

## 5. Novel Trends

In spite of the growing research interest in CNTs filler to reinforce polymer matrix and create sensitive pressure and strain sensors, many other researchers are going to realization of hybrid composites by combining other particles with carbon nanotube in order to add other features to the sensor. In fact, using two geometrically different fillers, such as graphene or graphene oxide with carbon nanotubes or metallic nanoparticles with CNTs, can have several advantages such as reduction of the fillers concentration as result of the formation of a co-supporting network of the two fillers which results a boosted electrical property as well mechanical and thermal properties at very low concentration. 

Furthermore, the combination of different nanoparticles is used to inhibit several effects such as temperature effects. As known, semiconductor CNTs possess negative thermal coefficient of resistance while metallic graphite has positive thermal coefficient of resistance. By the hybridization using tuned ratio of CNTs to graphite, a zero temperature coefficient of resistance (TCR) can be achieved as demonstrated in the work Luo et Liu [106]. In this work, they have succeeded to fabricate a sensitive piezoresistive strain sensor with self-temperature compensation. For the same objective, Ramalingame et al. [107] developed hybrid nanocomposite based on graphene and CNTs having nearly temperature-independent characteristic and with a sensitivity of 0.022 Ω/°C as shown in Figure 13. Amjadi and Sitti [108] proved that by hybridization, the possibility to have a highly stable and sensitive strain sensor accompanied by a broader temperature self-compensation range up to 100 °C in addition to retain the great temperature self-compensation under different relative humidity (RH) levels.

Furthermore, the already proven potential of using conductive polymer/CNT filaments to 3D-print parts could revolutionize the way sensing components are made by pushing forward the boundaries of current possibilities. As some recent works have suggested, applications for 3D printing technology using composites can be implemented for printing flexible tactile sensors as smart skin [109], components for sensitive platforms for explosives detection [110], printing of highly customized 3D structures for energy storage devices [111] and wearable energy storage [112] which represents a leap forward in wearable technology. In similar direction but implementing a different method of printing, Chavez et al. [113] reported that photopolymerizable nanocomposites made of commercially available photopolymers (e.g., Genesis) filled with MWCNT dispersions can be used for tunning the electrical and mechanical properties of 3D printed components in a commercial 3D printer (Form1+; Formlabs Inc., Somerville, MA, USA) by externally applying customized electric fields during the printing process. Finally, taking a step further in terms of sensing capabilities 3D printing will broaden the possibilities of sensing beyond the current boundaries. Enabling the fabrication of highly customizable sensors capable of sensing excitations beyond the here addressed sensors, as 3D printed biological sensors for replacing human senses as hearing or scaffolds/microfluidic structures sensitive to the presence of microbial activity [114]. In a very recent work, Ryan et al. [115] suggested the possibility to 3D print materials capable to be responsive to physical and chemical stimuli such as magnetic and electric fields, heat, light, pH and humidity.

Apart from 3D printed sensors, there is also a lot of progress in the direction of multifunctional wearable optical sensor based on an optical micro-/nanofibre to sense several stimuli such as physical and environmental quantities. In [116], Pan et al. developed a stretchable skin-like wearable optical sensor (SLWOS) embedded with a wavy optical micro/nanofibre (MNF) to form a highly sensitive sensor with a gauge factor of 675 at <1% strain. The flexible SLWOS endows the strain sensor also temperature sensing capability in the range of −20 to 130 °C due to the presence of PDMS that has a large negative thermo-optic coefficient enabling real-time monitoring of respiration, arm motion, and body temperature. Distinguishing several quantities such as bending, stretching, pressure was also achieved by Bai et al. [117] using dual-core elastomeric optical fibers.

## 6. Conclusions

In this paper, we discuss advancements of polymer/CNTs nanocomposites for strain and pressure sensors and focus on the sensing principles as well the influence of fabrication processes and resulting sensor properties. We cover thereby different structures for strain sensors based on polymer/CNTs including strain sensors based on randomly oriented CNTs, aligned CNTs and sandwiched type. The reported results show that electrical properties of the polymer/CNTs nanocomposites, which build the basis for the sensor properties, are dependent on multiple factors such as the CNTs type, size and orientation within the polymer matrix, the polymer viscosity, in addition to the processing conditions e. g. the mixing speed and temperature. These parameters affect enormously the conductive network formation within the nanocomposite and influence thereby both nominal resistance and impedance value as well as its sensitivity to mechanical forces. The results of several investigations show that high-performance polymer/CNTs based strain sensors have been developed and demonstrate a wide workable strain range up to 600%, high sensitivity up to 117,213, and good repeatability up to 180,000 cycles. This is strongly beyond state of the art of classical metal based and silicon-based sensors and demonstrate therefore the strength of this technological approach, which comes in addition to the fact that it is low-cost.

The sensors can be shaped as a scalable film, but also as filament, which allows them to be woven or embroidered into textiles and measure directly strain in a well-defined direction. These outstanding possibilities prove the potential of these stretchable strain sensors for integration in several applications including wearables, robotics and prosthetics. 

This paper illustrates the different configurations of polymer/CNTs pressure sensors based on the piezocapacitive principle, which are parallel plate structures with conductive dielectric layer and Interdigital electrode structure with conductive sensing layer. Although most sensors are evaluating the piezoresistive principle, pressure sensors based on polymer/CNTs show also an interesting piezocapacitive effect, which has an interesting behavior. This opens the possibility for an impedance measurement for the sensor.

Because of the outstanding properties of the CNTs in term of electrical, thermal properties. Nanocomposite materials show the potential to realize sensors for further measurement quantities such as temperature, humidity, gazes and IR illumination. Polymer/CNTs nanocomposite can also realize multifunctional sensors, so that sensors can measure more than one quantity at the same time. Despite the interesting properties of polymer/CNT nanocomposites, the cross-sensitivity can lead to instability and annoying influences on the sensor response. Therefore, recent developments are going in direction of self-compensation of the sensors by hybridization of the composite with other nanoparticles such as graphene, graphite and metallic nanoparticles.

In order to realize low-cost sensors based on polymer/CNTs nanocomposites with tunned mechanical and electrical properties, efforts are going to manufacturing using the 3D printing method.

## Figures and Tables

**Figure 1 sensors-21-00341-f001:**
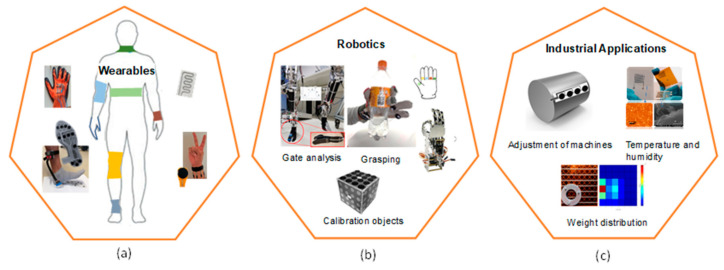
Examples of applications of strain and pressure sensor based stretchable and flexible nanocomposites: (**a**) wearables for human activity tracking and medical applications, (**b**) robotics application for grasping, humanoid robot gait analysis and calibration objects and (**c**) industrial applications.

**Figure 2 sensors-21-00341-f002:**
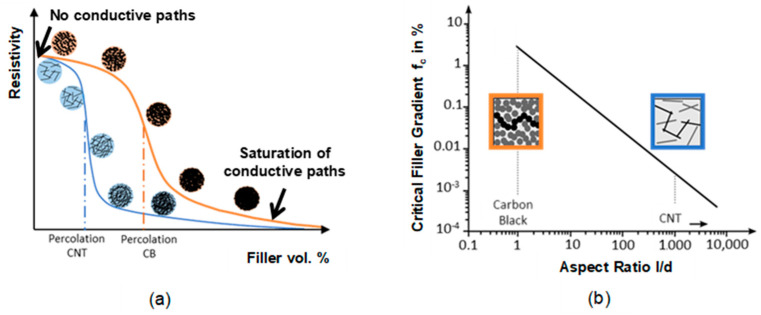
Influence of filler type: (**a**) composite resistivity behavior in dependence on filler volume concentration of carbon nanotubes (CNTs) and carbon black (CB), (**b**) critical filler gradient in dependence on the aspect ratio.

**Figure 3 sensors-21-00341-f003:**
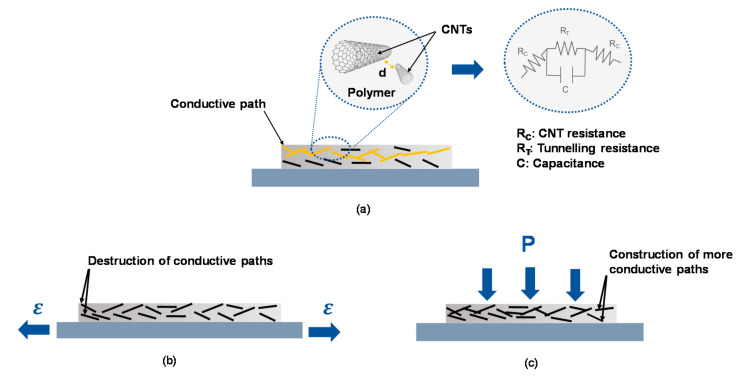
Conduction mechanisms of polymer/carbon nanotube (CNT) sensor: (**a**) without load, (**b**) under strain and (**c**) under pressure.

**Figure 4 sensors-21-00341-f004:**
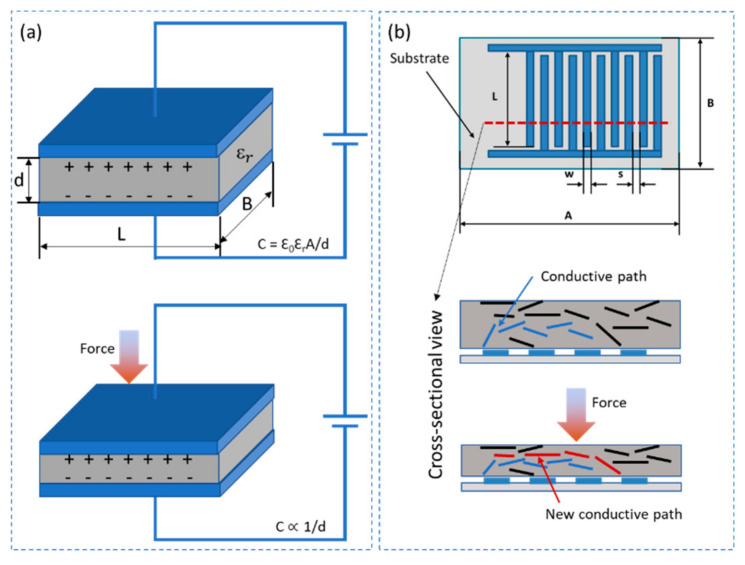
Conduction mechanisms of polymer/CNT pressure sensor based on piezocapacitive working principle: (**a**) sensors based on parallel plate electrode structures and (**b**) sensors based on interdigital electrode structures.

**Figure 5 sensors-21-00341-f005:**
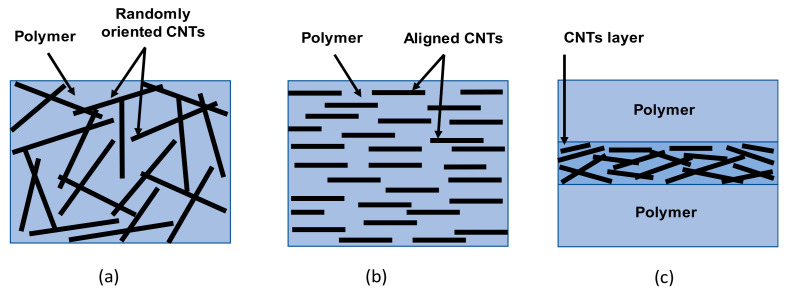
Different polymer/CNTs architectures: (**a**) randomly oriented polymer/CNTs nanocomposite, (**b**) aligned CNTs in polymer matrix and (**c**) sandwiched CNTs layers between two layers of polymer.

**Figure 6 sensors-21-00341-f006:**
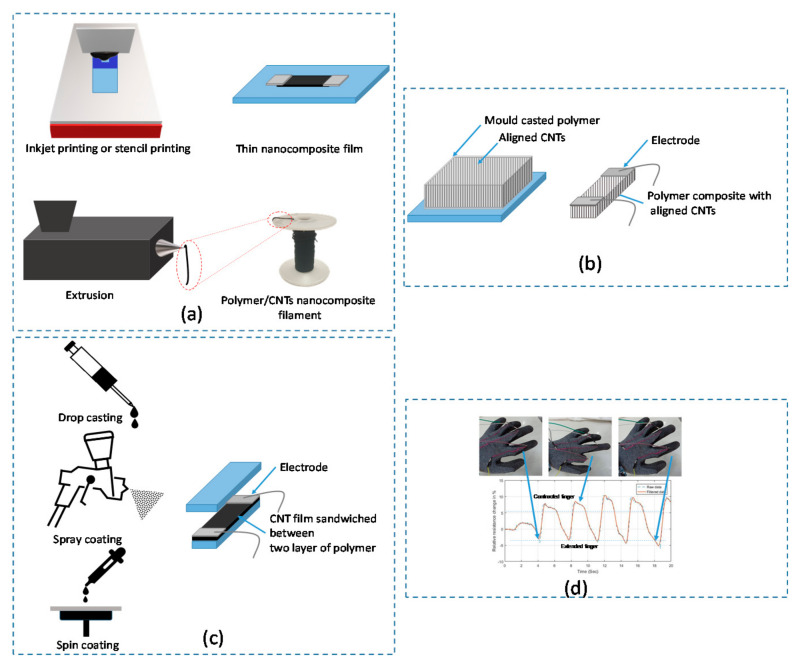
Different sensors architecture, deposition techniques and potential application (**a**) Strain sensor based on randomly oriented CNTs, (**b**) sandwiched CNTs based strain sensor (**c**) strain sensor based on aligned CNTs and (**d**) potential of filament strain sensors for wearable application.

**Figure 7 sensors-21-00341-f007:**
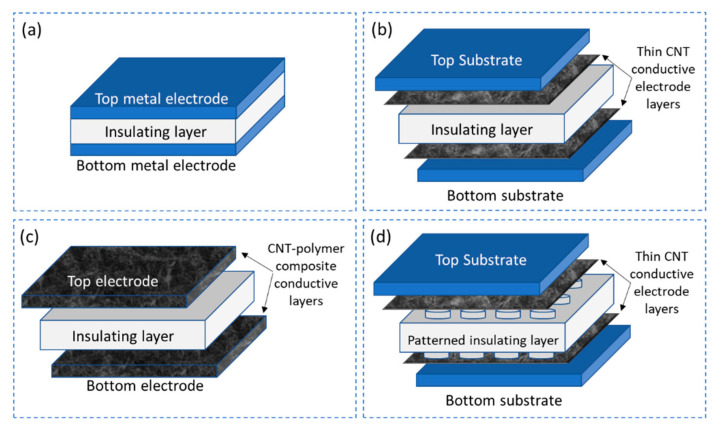
The parallel plate capacitive pressure sensor with insulation dielectric layer can be categorized in certain forms (**a**) generic form with metallic electrodes, (**b**) thin CNT layers deposited on the substrate acting as electrodes, (**c**) CNT-polymer composite based conductive electrode layers and (**d**) patterning the insulating layer to enhance the performance.

**Figure 8 sensors-21-00341-f008:**
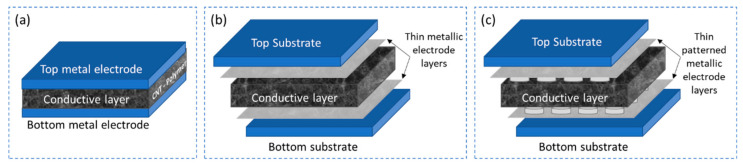
The parallel plate capacitive pressure sensor with conductive sensing layer can be categorized in certain forms (**a**) generic form with metallic electrodes, (**b**) thin metallic layers deposited on the substrate acting as electrodes and (**c**) patterning the metallic layer to enhance the performance.

**Figure 9 sensors-21-00341-f009:**
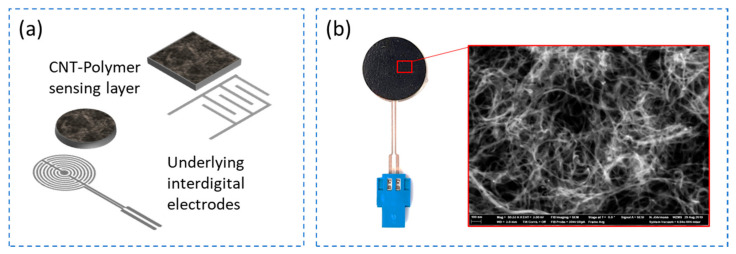
(**a**) Different interdigital electrode structure with conductive sensing layer on top and (**b**) CNT-PDMS based pressure sensor with circular sensing and interdigital electrode layers and an SEM image representing the distribution of CNT network in the polymer.

**Figure 10 sensors-21-00341-f010:**
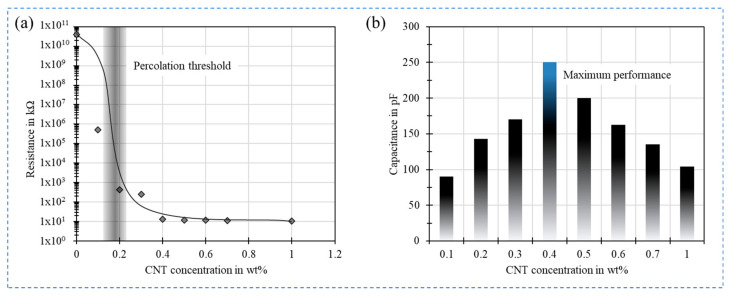
Effect of CNT concentration on the performance of the sensor: (**a**) piezoresistive sensors and (**b**) piezocapacitive sensors.

**Figure 11 sensors-21-00341-f011:**
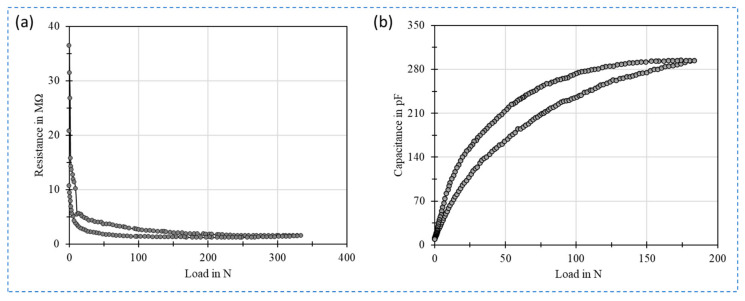
(**a**) Piezoresistive behavior of the sensor and (**b**) piezocapacitive behavior of the sensor.

**Figure 12 sensors-21-00341-f012:**
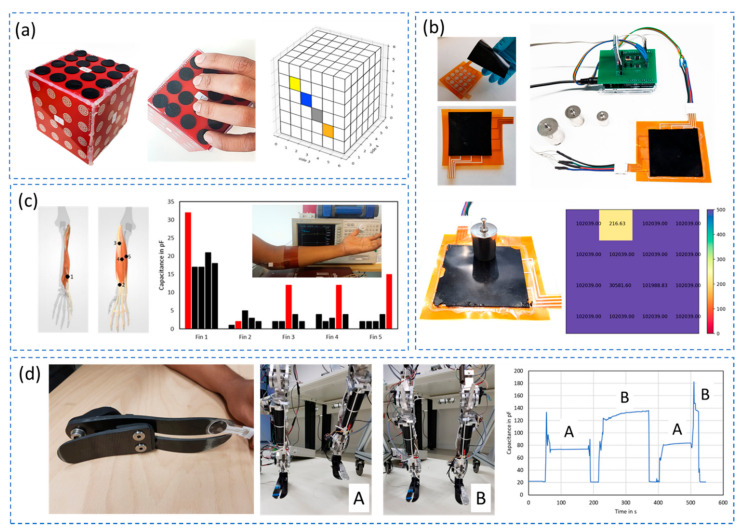
Potential application of nanocomposite pressure sensors: (**a**) touch and grasping, (**b**) matrix for force distribution, (**c**) body attached for muscle movement monitoring and (**d**) robot gait analysis.

**Figure 13 sensors-21-00341-f013:**
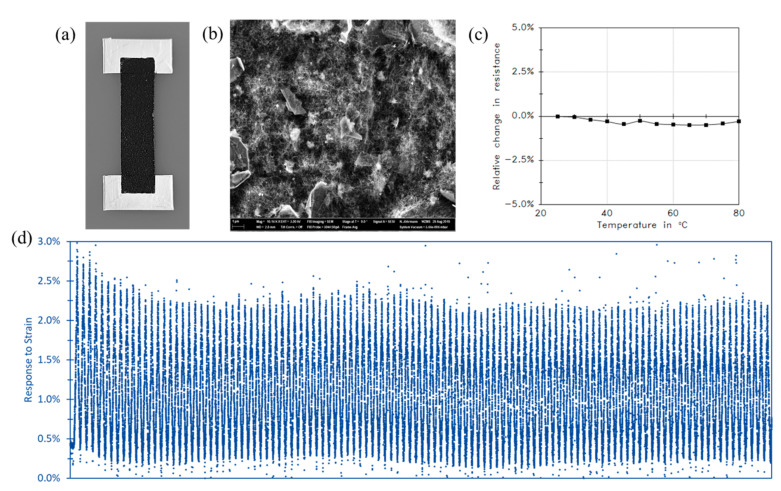
Thin film CNT-graphene hybrid nanocomposite-based temperature self-compensated strain sensor. (**a**) Photograph of the hybrid nanocomposite, (**b**) SEM image of the CNT-graphene nanocomposite, (**c**) relative change in resistance of the CNT-graphene hybrid nanocomposite between 20 and 80°C and (**d**) Response of the hybrid sample under cyclic strain loading test.

**Table 1 sensors-21-00341-t001:** Influence of processing conditions on the electrical properties.

Polymer Matrix	CNTs Filler	Aspect Ratio	Fabrication Process	Electrical Conductivity/Resistivity	Percolation Threshold	Ref.
Polyester	MWCNTs	D = 50 nm, L = 10–20 μm	Stirring	3.9 × 10^−2^ (Ω m)^−1^	0.6%	[26]
Polyurethane	Amino-CNT	D = 8–15 nm, 50 nm	Co–coagulation and compression molding method	<4.48 × 10^−6^ S cm^−1^	0.337 vol%	[27]
Epoxy	SWCNTs	D = 1.34 nm, L = 1–10 μm	Solvent processing in acetone	~4.74 × 10^6^ S cm^−1^	Between 0.2 and 0.5 wt.%	[28]
Epoxy	Pristine XD CNT	-	Solvent processing in ethanol	-	weight fraction of 0.05 and 0.1 wt.%	[28]
Polystyrene (PS)	MWCNTs	-	Solution mixing in toluene	-	6 wt.% MWCNT	[29]
Polyvinylidene-fluoride (PVDF)	MWCNTs	-	Melt-mixing	219,000 Ω	between 1 and 1.25 wt.% MWCNTs	[30]
PMMA	MWCNTs	ID = 5–10 nm, OD = 60–100 nm, L = 0.5– 500 μm	Solvent mixing in chloroform	1.6–107 Ω/sq	0.8–1%	[31]
PEO	MWCNTs	-	Coagulation	-	Between 0.5 wt.%–1 wt.%	[32]
Polyurethane-urea (TPU)	amino-functionalized MWNTs	-	Solution processing	-	0.35 wt.%	[33]

**Table 3 sensors-21-00341-t003:** Previous research work on pressure sensors. PP—parallel plate structure, IDE—interdigital electrode structure, PR—piezoresistive, PC—piezocapacitive.

Sensing Layer	Electrode Material	CNT Concentration	Pressure Range	Sensitivity	Sensor Structure	Working Principle	Ref.
PDMS-CNT	ITO	7 wt.%	0–3.5 kPa	8.3 kPa^−1^	PP	PR	[12]
PDMS-MWCNT	Aluminum	1 wt.%	0.45 MPa–1.2 MPa	-	PP	PR	[10]
PDMS-MWCNT	Chromium-Silver	8 wt.%	0–1.4 N	-	IDE	PR	[102]
PDMS-carbon flakes	Silver	25%, 50%, 75%	0.5 N–10 N	-	IDE	PR	[103]
PDMS-CNT	Copper	15 wt.%	1.25 kPa–43.75 kPa	-	IDE	PR	[104]
Ecoflex	CNT-PDMS	10 wt.%	50 kPa–1.2 Mpa	-	PP	PC	[95]
Microstructured PDMS	CNT	0.3 wt.%	10 mN–1 N	20%	PP	PC	[13]
Parylene	MWCNT/Gold	-	0–758 Pa	1.33 kPa^−1^	PP	PC	[93]
PDMS	CNT ink	-	15.2 kPa–337 kPa	0.021% kPa^−1^	PP	PC	[94]
Liquid Crystal-CNT	Gold	0.5 wt.%	0–400 kPa	-	PP	PC	[98]
PDMS-CNT	Ion gel	1.5 wt.%	0–8 kPa	9.55 kPa^−1^	PP	PC	[99]
PDMS-MWCNT	Silver	1 wt.%	10 mN–180 N	25%/N in 0–0.01 N 46.8%/N in 0–1 N	IDE	PC	[40]

## Data Availability

Data sharing not applicable.

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
