# Peer review of "Review on Conductive Polymer/CNTs Nanocomposites Based Flexible and Stretchable Strain and Pressure Sensors"

_sensors, 2021, doi:10.3390/s21020341_

Round 1
Reviewer 1 Report
Kanoun et al presented a detailed review about the flexible and stretchable strain and pressure sensors based on CNTs nanocomposites. I believe this review is of interest to the broad readership of Sensors. However, this manuscript was poorly prepared. I recommended its publication after major revision indicated below.
- In introduction, the authors mentioned this paper is divided into 6 main sections mentioned, and Section 5 summarizes different applications of the nanocomposite and illustrates materials performance in addition to their limitations. Section 6 includes possible future developments towards flexible strain and pressure sensors. However, in the manuscript, section 5 is Novel Trends and section 6 is Conclusions. Please make the tile and contents consistent in the whole manuscript. In addition, I suggest the authors include optical sensing approaches in section 5. For example, optical micro/nanofiber based strain sensor (Nanoscale, 2020, 12, 17538) and Stretchable distributed fiber-optic sensors. (Science 2020, 370, 848.)
- The author introduced the piezocapacitive effect in section 2.3 but did not mention its mechanism. Instead, the interdigital electrode was presented at length. However, the interdigital electrode should be related to the structure design of pressure sensors. Thus, the author had better introduce the principle of piezocapacitive pressure sensors based on conductive fillers/polymer composites.
- The author mentioned the sonication and its two method, but did not introduce them in detail. The sonication is a very effective method to disperse CNTs, especially the sonication tip. The author should compare two of them, because the sonication approach is mentioned many times in the following references.
- The author presented many opinions, but some of them have no reference to support. For example, there is no reference to support the In-situ polymerization.
- The author described the disadvantages of non-covalent functionalization of CNTs, without mention of its advantages. The author should also present the advantages.
- On page 9, what is the “34” in the sentence “34 Development of flexible and stretchable strain and pressure sensors based on polymer/ CNTs nanocomposite?
- On page 13, where is the figure 7c mentioned on line 455?
- The author introduced the randomly oriented CNTs composites, flexible polymer composite with oriented CNTs and sandwiched structures. However, what is the advantage of each type of composite, the review should compare their difference and summary it.
- The author stated that” capacitive strain sensors based on polymer/CNTs require integration of more CNTs within the nanocomposite materials which leads to non-cost-effective solutions.” However, in following sections (4.2), the author introduced some corresponding capacitive sensors (79, 80) which are not mentioned in Table 2. In addition, the author should list the reference to support that “capacitive strain sensors based on polymer/CNTs require integration of more CNTs within the nanocomposite materials.”
- On line 594, where is the figure 8e, and line 613, where is the figure 8g?
- On line 620, the figure 10a does not present a sensor? It seems to be numbered wrong.
- The author does not mention the references on structure design of the CNTs/Polymer composite, like pyramid, porous. it is significant to endow these structures to sensors for performance improvement.
Author Response
Kanoun et al presented a detailed review about the flexible and stretchable strain and pressure sensors based on CNTs nanocomposites. I believe this review is of interest to the broad readership of Sensors. However, this manuscript was poorly prepared. I recommended its publication after major revision indicated below. 
- In introduction, the authors mentioned this paper is divided into 6 main sections mentioned, and Section 5 summarizes different applications of the nanocomposite and illustrates materials performance in addition to their limitations. Section 6 includes possible future developments towards flexible strain and pressure sensors. However, in the manuscript, section 5 is Novel Trends and section 6 is Conclusions. Please make the tile and contents consistent in the whole manuscript. In addition, I suggest the authors include optical sensing approaches in section 5. For example, optical micro/nanofiber based strain sensor (Nanoscale, 2020, 12, 17538) and Stretchable distributed fiber-optic sensors. (Science 2020, 370, 848.)
The reviewer has right. We have changed the numerotation of section in the text and we have included some references related to optical micro/nanofiber-based strain sensor.
- The author introduced the piezocapacitive effect in section 2.3 but did not mention its mechanism. Instead, the interdigital electrode was presented at length. However, the interdigital electrode should be related to the structure design of pressure sensors. Thus, the author had better introduce the principle of piezocapacitive pressure sensors based on conductive fillers/polymer composites.
The concept of piezo capacitive effect is explained by the structure difference of the electrode arrangement. Both parallel plate and interdigital structure contribute to the realization of piezo capacitive effect and this is well highlighted in this article.
- The author mentioned the sonication and its two method, but did not introduce them in detail. The sonication is a very effective method to disperse CNTs, especially the sonication tip. The author should compare two of them, because the sonication approach is mentioned many times in the following references.
More details about the difference between tip sonicator and bath sonicator have been included.
- The author presented many opinions, but some of them have no reference to support. For example, there is no reference to support the In-situ polymerization.
Thank you for pointing that out, new references have been added in many parts of the manuscript, and in particular for the in-situ polymerization.
- The author described the disadvantages of non-covalent functionalization of CNTs, without mention of its advantages. The author should also present the advantages.
Non-covalent functionalization is attaching the functional groups on CNT based on supramolecular complexation using various adsorption forces, such as Van der Waals force, hydrogen bonds, electrostatic force and π-stacking interactions, compared to other methods it could be carried out on the relatively mild reactions conditions.
- On page 9, what is the “34” in the sentence “34 Development of flexible and stretchable strain and pressure sensors based on polymer/ CNTs nanocomposite?
The reviewer is right. It has been changed to “4. Development of flexible and stretchable strain and pressure sensors based on polymer/ CNTs nanocomposite”.
- On page 13, where is the figure 7c mentioned on line 455?
Figure 7c indicates the subfigure (c) in the figure 7. Now all the figures caption has been changed e.g., Figure 7c to Figure 7 (c).
- The author introduced the randomly oriented CNTs composites, flexible polymer composite with oriented CNTs and sandwiched structures. However, what is the advantage of each type of composite, the review should compare their difference and summary it.
The reviewer is right. A comparison between the different structure of strain sensors is missing. We have included mored details in the section 4.1.2.
- The author stated that” capacitive strain sensors based on polymer/CNTs require integration of more CNTs within the nanocomposite materials which leads to non-cost-effective solutions.” However, in following sections (4.2), the author introduced some corresponding capacitive sensors (79, 80) which are not mentioned in Table 2. In addition, the author should list the reference to support that “capacitive strain sensors based on polymer/CNTs require integration of more CNTs within the nanocomposite materials.”
The section 4.1 focus on strain sensor, while section 4.2 is on pressure sensor. Capacitive principle is interesting for pressure sensor as the sensitivity is higher at lower CNTs concentration leading to large ΔC/C0 when pressure is increased. However, in case of strain measurement, higher concentration is required to achieve larger ΔC change
- On line 594, where is the figure 8e, and line 613, where is the figure 8g?
It has been corrected. Figure 8e changed to Figure 8b and Figure 8g is changed to 8c.
- On line 620, the figure 10a does not present a sensor? It seems to be numbered wrong.
Figure 10a is corrected as figure 9a
- The author does not mention the references on structure design of the CNTs/Polymer composite, like pyramid, porous. it is significant to endow these structures to sensors for performance improvement.
The aspect of the structuring of the dielectric layer, nanocomposite layer and electrodes for the performance enhancement of the sensor is well addressed in the article and the corresponding references of interest are [101] - nanoporous layer; [103] - pyramid electrodes.

Reviewer 2 Report
In the article entitled Review on Conductive Polymer/CNTs Nanocomposites based Flexible and Stretchable Strain and Pressure Sensors, the sensors currently used in various applications were reviewed, including in robotics, prosthetics etc. Selected groups of sensors were characterized, and the influence of their production process on the achieved sensory properties was discussed.
Fig. 1 and Fig. 2 are very illegible. They need to be corrected.
Figure captions should be standardized - use the same font size.
Section: Development of flexible and stretchable strain and pressure sensors based on polymer / CNTs nanocomposite should be number 4, not 34. Please change it.
The wrong numbering of sections: 1 and 2 on pages 10 and 11 introduces some graphic chaos. Please change it.
Please check the numbers of figures and tables throughout the text. The order of citing figures and tables in the text also needs to be checked. Their order is incorrect, e.g. on page 5 there is figure 3, not figure 1, on page 6 there is figure 4 and not figure 2.
The figures included were not described sufficiently in the text. This should be corrected.
It is difficult for me to refer to the topic itself, but as a rule it is difficult to find an innovative approach to the topic in review articles. The only novelty in this article are the described prospects for sensor production and their new applications. The introduction section requires a more detailed description of the current state of knowledge related to the topic. The aim of the work was not specified in detail either. What is the purpose of summarizing all the sections in the first part of the article?
I recommend this article for publication after making all the changes described above.
Author Response
In the article entitled Review on Conductive Polymer/CNTs Nanocomposites based Flexible and Stretchable Strain and Pressure Sensors, the sensors currently used in various applications were reviewed, including in robotics, prosthetics etc. Selected groups of sensors were characterized, and the influence of their production process on the achieved sensory properties was discussed.
Fig. 1 and Fig. 2 are very illegible. They need to be corrected.
Figure captions should be standardized - use the same font size.
We have adjusted the clarity of Fig.1 and Fig. 2.
Font size has been standardized
Section: Development of flexible and stretchable strain and pressure sensors based on polymer / CNTs nanocomposite should be number 4, not 34. Please change it.
It has been changed.
The wrong numbering of sections: 1 and 2 on pages 10 and 11 introduces some graphic chaos. Please change it.
Two “Figures 1” are presented which has been changed.
DHIVAKAR: Please check the numbers of figures and tables throughout the text. The order of citing figures and tables in the text also needs to be checked. Their order is incorrect, e.g. on page 5 there is figure 3, not figure 1, on page 6 there is figure 4 and not figure 2.
Order of citing figures and tables has been checked and corrected
Following figure has been cited after correction: Figure 3, Figure 4, Figure 5, Table 1, Figure 6c, Figure 12a
The figures included were not described sufficiently in the text. This should be corrected.
This has been corrected.
It is difficult for me to refer to the topic itself, but as a rule it is difficult to find an innovative approach to the topic in review articles. The only novelty in this article are the described prospects for sensor production and their new applications. The introduction section requires a more detailed description of the current state of knowledge related to the topic. The aim of the work was not specified in detail either. What is the purpose of summarizing all the sections in the first part of the article?
We have included more details in the introduction and illustrated the novel aspects of this review paper.
I recommend this article for publication after making all the changes described above.

Reviewer 3 Report
The review by Kanoun et al. presents an important contribution on the fabrication of polymer/CNT nanocomposites for flexible and stretchable strain and pressure sensors. In my opinion, the review is complete is the description of different protocols to obtain and apply such nanocomposites. I have some few suggestions/corrections for the authors improve the final version:
1- Give additional explanation of Figure 2; it is no clear which are the best conditions when using CNT or CB composites. This figure can be placed in the second section.
2- Two “Figures 1” are presented. All figures need to be renumbered.
3- The authors presented a section devoted to filament composite. As these filaments are potentially applied for 3D printing, I suggest including some examples if there are. If not, the authors may include in the trends section as 3D printing can be used to fabricate flexible sensors using such filaments, as previous works have shown for other sensing applications such as:
3D Printing Technologies for Flexible Tactile Sensors toward Wearable Electronics and Electronic Skin, Polymers 2018, 10(6), 629; https://doi.org/10.3390/polym10060629
3D-printed flexible device combining sampling and detection of explosives, Sensors and Actuators B: Chemical, Volume 292, 1 August 2019, Pages 308-313
A 3D-printed stretchable structural supercapacitor with active stretchability/flexibility and remarkable volumetric capacitance, Journal of Materials Chemistry A, https://doi.org/10.1039/D0TA04460A
3D printing of highly flexible supercapacitor designed for wearable energy storage, Materials Science and Engineering: B, Volume 226, December 2017, Pages 29-38
4- Is there any example of CNT/polymer fabricated by UV curable prototyping or 3D printing? This may be an interesting future method of fabrication. The following reference reports an interesting protocol.
Electrical and mechanical tuning of 3D printed photopolymer–MWCNT nanocomposites through in situ dispersion, Journal of Applied Polymer Science, Volume 136, Issue 22, 2019
5- These polymer/CNTs composites present huge applicability for electrochemical sensing, although this application is not the topic of the review. So, I suggest including a brief mention in conclusions or novel trends the potential application of these materials in other wearable sensing areas, such as electrochemical sensing. Examples of some reviews on potential applications for polymer/CNT materials as follows:
Future of additive manufacturing: Overview of 4D and 3D printed smart and advanced materials and their applications, Chemical Engineering Journal 403 (2021) 126162
3D-printed sensors: Current progress and future challenges, Sensors and Actuators A: Physical, Volume 305, 15 April 2020, 111916
Author Response
The review by Kanoun et al. presents an important contribution on the fabrication of polymer/CNT nanocomposites for flexible and stretchable strain and pressure sensors.  In my opinion, the review is complete is the description of different protocols to obtain and apply such nanocomposites. I have some few suggestions/corrections for the authors improve the final version:
1- Give additional explanation of Figure 2; it is no clear which are the best conditions when using CNT or CB composites. This figure can be placed in the second section.
We have included more details related to figure 2.
2- Two “Figures 1” are presented. All figures need to be renumbered.
It has been re-numbered
3- The authors presented a section devoted to filament composite. As these filaments are potentially applied for 3D printing, I suggest including some examples if there are. If not, the authors may include in the trends section as 3D printing can be used to fabricate flexible sensors using such filaments, as previous works have shown for other sensing applications such as:
- 3D Printing Technologies for Flexible Tactile Sensors toward Wearable Electronics and Electronic Skin, Polymers 2018, 10(6), 629; https://doi.org/10.3390/polym10060629
- 3D-printed flexible device combining sampling and detection of explosives, Sensors and Actuators B: Chemical, Volume 292, 1 August 2019, Pages 308-313
- A 3D-printed stretchable structural supercapacitor with active stretchability/flexibility and remarkable volumetric capacitance, Journal of Materials Chemistry A, https://doi.org/10.1039/D0TA04460A
- 3D printing of highly flexible supercapacitor designed for wearable energy storage, Materials Science and Engineering: B, Volume 226, December 2017, Pages 29-38
This is a very good observation and you are right; these filaments could be used for 3D printing. A paragraph precisely in this direction was added to the filament section, in which publications we found about CNT-polymer filaments used for 3D printing were added. Plus, we included the publications you kindly suggested as well in the trend section. Many thanks for the remark.
 4- Is there any example of CNT/polymer fabricated by UV curable prototyping or 3D printing? This may be an interesting future method of fabrication. The following reference reports an interesting protocol.
- Electrical and mechanical tuning of 3D printed photopolymer–MWCNT nanocomposites through in situ dispersion, Journal of Applied Polymer Science, Volume 136, Issue 22, 2019
Thank you for the insight, we were not aware of this possibility. This point was addressed in the Novel Trends section and the reference you recommended was included.
5- These polymer/CNTs composites present huge applicability for electrochemical sensing, although this application is not the topic of the review. So, I suggest including a brief mention in conclusions or novel trends the potential application of these materials in other wearable sensing areas, such as electrochemical sensing. Examples of some reviews on potential applications for polymer/CNT materials as follows:
- Future of additive manufacturing: Overview of 4D and 3D printed smart and advanced materials and their applications, Chemical Engineering Journal 403 (2021) 126162
- 3D-printed sensors: Current progress and future challenges, Sensors and Actuators A: Physical, Volume 305, 15 April 2020, 111916
Electrochemical sensors are indeed not the focus of this review. However, we certainly see the connection between the two topics in the direction of wearable sensors. We added a brief mention of this possibility at the end of the Novel trend section. Thank you for the remark.
Round 2
Reviewer 1 Report
The authors have addressed all my concerns, I recommended its
publication.